# Understanding reasons for suboptimal tuberculosis screening in a low-resource setting: A mixed-methods study in the Kingdom of Lesotho

Afom T. Andom[1,2]*, Hannah N. Gilbert[2], Melino Ndayizigiye[1], Joia S. Mukherjee[2,3,4], Jonase Nthunya[1], Tholoana A. Marole[1], Mary C. Smith Fawzi[2], Courtney M. Yuen[2,3]

1 Partners In Health–Lesotho, Maseru, Lesotho, 2 Department of Global Health and Social Medicine, Harvard Medical School, Boston, MA, United States of America, 3 Division of Global Health Equity, Brigham and Women's Hospital, Boston, MA, United States of America, 4 Partners in Health, Boston, MA, United States of America

* aandom@pih.org

## Abstract

Lesotho has one of the highest tuberculosis (TB) incidence rates in the world, estimated at 654/100,000 population. However, TB detection remains low, with only 51% of people with TB being diagnosed and treated. The aim of this study was to evaluate implementation of TB screening and identify drivers of suboptimal TB screening in Lesotho. We used a convergent mixed methods study design. We collected data on the number of health facility visits and the number of clients screened for TB during March-August, 2019 from one district hospital and one health center. We conducted interviews and focus group discussions with patients and health workers to elucidate the mechanisms associated with suboptimal screening. Out of an estimated 70,393 visitors to the two health facilities, only 22% of hospital visitors and 48% of health center visitors were asked about TB symptoms. Only 2% of those screened at each facility said that they had TB symptoms, comprising a total of 510 people. Lack of training on tuberculosis screening, overall staff shortages, barriers faced by patients in accessing care, and health care worker mistrust of tuberculosis screening procedures were identified as drivers of suboptimal TB screening. TB screening could be improved by ensuring the availability of well-trained, incentivized, and dedicated screeners at health facilities, and by providing TB screening services in community settings.

## Introduction

Globally, around 30 percent of people with tuberculosis (TB) are neither diagnosed nor treated [1]. Timely detection of TB and immediate initiation of effective treatment are important to save lives, reduce morbidity from the disease, and control transmission [2, 3]. In settings with high TB burdens, actively screening key populations for TB is a critical step for closing the TB diagnosis gap [4]. In settings with high TB burdens, one population that can benefit from

**Data Availability Statement:** All data supporting the findings of this study are available within the article and supplementary materials.

**Funding:** This work was conducted with support from the Master of Medical Sciences in Global Health Delivery program of Harvard Medical School Department of Global Health and Social Medicine (ATA) and financial contributions from Harvard University and the Ronda Stryker and William Johnston MMSc Fellowship in Global Health Delivery (ATA). The content is solely the responsibility of the authors and does not necessarily represent the official views of Harvard University and its affiliated academic health care centers. Additional support was provided by Partners In Health – Lesotho (ATA). The funders had no role in study design, data collection and analysis, decision to publish, or preparation of the manuscript.

**Competing interests:** The authors declare no competing interests.

systematic TB screening is people attending health facilities. In settings with high HIV prevalence, health facility attendees include large numbers of people living with HIV, making facility-based screening particularly effective for TB detection. Prevalence surveys in sub-Saharan African countries have repeatedly found that many people with undiagnosed TB had in fact sought care for their symptoms [5]. Screening health facility attendees has been shown to increase overall TB diagnoses in the population in countries with high TB and HIV burdens [6, 7]. However, even though countries may have policies for screening health facility attendees, incomplete implementation leads to missed diagnoses [8, 9].

Lesotho has the highest TB incidence globally, with an estimated 654 cases per 100,000 population annually [1]. The prevalence of multidrug resistant TB (MDR-TB) is among the highest in sub-Saharan Africa, and HIV is a major driver of the TB epidemic given the adult HIV prevalence of 21% [10]. In Lesotho, the estimated TB detection rate is only 51%, meaning that approximately half of TB patients are neither diagnosed nor treated [1]. The high prevalence of HIV co-infection among TB patients (62%) [1] is a challenge for TB diagnosis since people living with HIV often have paucibacillary or extrapulmonary disease that is less likely to be detected by sputum testing [11]. In Lesotho, there is no routine TB active case-finding outside health facilities, but screening of health facility clients is indicated in the national guidelines [12].

With this high rate of TB in the Kingdom, it is vital that every person presenting for care for whatever reason be screened for TB symptoms, tested if symptoms are present and given treatment if diagnosed. We therefore conducted a mixed methods study to evaluate TB screening and understand the reasons for suboptimal screening in two health facilities in Berea district in the Kingdom of Lesotho.

## Materials and methods

### Ethics statement

This study was approved by the National Health Research Ethics Committee of the Kingdom of Lesotho (ID91-2020) and by Harvard Medical School Institutional Review Board (protocol: IRB20-0109). All people who participated in the interviews and focus group discussions provided written informed consent.

### Study design

We conducted a convergent mixed methods study [13] to assess the TB care cascade from screening through treatment completion in two health facilities in Lesotho. This paper focuses on screening, which is the initial step of the care cascade. Qualitative methods and results are reported according to Consolidated Criteria for Reporting Qualitative Research (COREQ) guidelines.

### Study setting

The study was conducted in Berea Hospital and Khubetsoana Health Center. Berea Hospital is a district-level referral hospital located approximately 40 km from Maseru, the capital city of Lesotho. Khubetsoana Health Center is a primary-level facility near the outskirts of Maseru. Both facilities offer TB diagnostic and treatment services.

Lesotho's national TB guidelines indicate that people presenting to health facilities with persistent cough or other TB symptoms should be registered as having "presumptive TB" and should be asked to submit sputum for testing [12]. The guidelines indicate that TB screeners (often lay people) should identify people who are coughing in health facility waiting areas, that

people living with HIV should be screened at every clinical encounter, and that TB screening should be routine in all health facility departments. All those with TB are initiated on treatment, which is available for free throughout the country.

## Quantitative data collection and analysis

We sought to assess the proportion of people who attended the two health facilities during March-August 2019 who were screened for TB. To define the target population, we estimated the total visits to each health center during the analytic period based on monthly reports to the district health management office, summing up the total visits made to the outpatient department, ART clinic, antenatal clinic, under-five clinic, and family planning clinic. We collected data on the number of clients screened for TB from the paper-based TB screening register, and we collected data on the number of clients with TB symptoms from the paper-based presumptive TB register. We manually counted the number of people entered into each register by month and entered the data as aggregate numbers into a Microsoft Excel spreadsheet for analysis. We calculated the percentage of people screened by dividing the number screened by the total estimated visits to the health facilities, assuming that each visit corresponded to a single visitor who should be screened (i.e., a person accessing health services for him/herself or a guardian bringing a child to access services).

## Qualitative data collection

Purposeful sampling was used to identify participants for qualitative data collection. To maximize variation, we included participants who had a wide range of experiences with TB care in Lesotho. The participants of both the interviews and focus group discussions (FGDs) were recruited with the help of the nurse managers in both health facilities and then, the research assistant explained the purpose of the study to the potential participants via a telephone call. The majority of the people contacted for the interviews and focus group discussions participated in the study, but we did not collect information on the number who refused.

Overall, there were 53 participants. We conducted a single round of semi-structured in-depth interviews with 15 health care workers; variation was ensured by purposefully sampling for a variety of professional backgrounds (district health manager = 1, TB program coordinators = 2, village health workers = 4, TB screeners = 4, laboratory personnel = 2, and implementing partners = 2). Two FGDs were conducted with nurses; each FGD included 7 participants. We also conducted a single round of semi-structured in-depth interviews with 24 patients.

All interviews and FGDs were conducted by local research assistants (JN, TAM), one of whom was male and one of whom was female. Both were university graduates with over 5 years of research experience and training in qualitative data collection methods. The research assistants did not previously know any of the participants. Interviews took place in a private room at the study health facilities, and were conducted in Sesotho or English, according to the participant's stated preference. Interviews and FGDs took place in a private room out of earshot of others in the selected health facilities. Prior to the interview or FGD, the research assistants provided a standard introduction to the purpose of the study. No one else was present besides the research assistants and participants. Interviews and FGDs lasted on average between 60 and 90 minutes and were audio-recorded with permission. Field notes were also taken by the research assistants to aid in the transcription process.

Interviews and FGDs followed semi-structured interview guides (S1 Text) that were developed for each population by the lead author (AA, a medical doctor with 4 years of experience working in Lesotho) and second author (HG, a medical anthropologist). The study guide was

piloted before the start of the data collection. Topics for health worker interviews and FGDs with nurses addressed training, supervision, supplies, and practices related to screening and diagnostic testing. The patient interview guide covered topics related to care-seeking experiences including access to services and treatment, as well as barriers and facilitators to receiving appropriate diagnosis and care. Because the study as a whole sought to assess the entire care cascade, interview guides explored different steps along the cascade, but this manuscript focuses only on the parts of the interviews relevant to screening. AA and HG regularly reviewed transcripts to monitor data quality and ensure fidelity to the research aims. Transcripts were not returned back to the participants for comments.

## Qualitative data analysis

We used an inductive, thematic content analysis approach to identify key concepts related to gaps in TB screening [14]. A subset of transcripts was open coded by AA to identify content related to barriers and facilitators to appropriate screening. HG reviewed the open coding results; discrepancies were resolved through discussion, and the final concepts were developed into a draft codebook. The codebook was piloted and revised by AA and HG; the final codebook consisted of 38 codes and was used to code the dataset using Dedoose version 8 qualitative data management software. Coded data were analyzed inductively by AA to identify an initial set of descriptive themes which were labeled, defined, and supported with excerpts from the data. The initial draft of the thematic categories was created by AA and reviewed by HG. Employing an iterative process, AA and HG examined links between initial thematic categories to develop a set of increasingly higher-level concepts; saturation was reached when no higher-level concepts emerged. This resulted in four comprehensive thematic categories that that were defined and elaborated by AA to explain low TB screening in health facilities (S1 Table). Findings were not shared with patient participants, but health system participants will have access to the findings when they are shared with the two participating health facilities and the TB program at the Ministry of Health.

## Results

### Screening coverage

District reports recorded 45,699 visits to Berea Hospital and 24,694 visits to Khubetsoana Health Center during the 6-month analysis period. Based on the TB screening register, we estimate that only 22% (n = 9,841) of hospital visitors and 48% (n = 11,840) of visitors at the health center were asked about TB symptoms. Only 2% (n = 218 in the hospital, n = 292 in the health facility) of these individuals were listed in the presumptive TB register as having reported symptoms.

### Barriers to effective TB screening

Our study's qualitative portion revealed four thematic categories that help explain the low percentage of visitors screened for TB and the low percentage reporting symptoms in Berea district. The thematic categories are: (1) Overall staff shortage in health facilities; (2) Lack of adequate training for screeners; (3) Structural barriers create delays that shape patients' care priorities and (4) Internal mistrust among health care workers about the accuracy of screening, which leads to redundant procedures. Fig 1 shows how these emergent themes intersect.

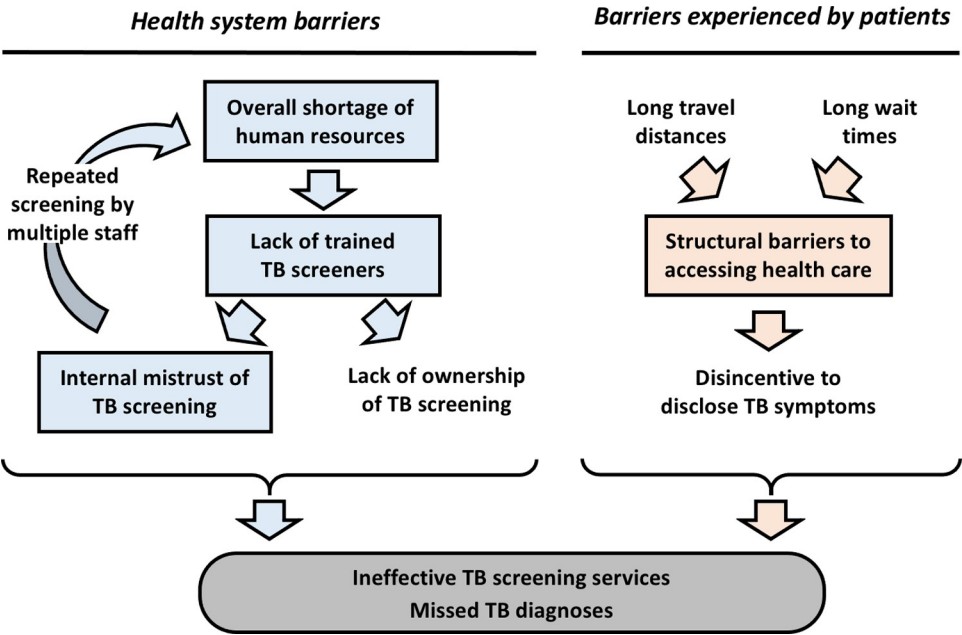

**Fig 1. Conceptual framework for how barriers to implementing TB screening in health facilities result in missed TB diagnoses.** Major thematic categories from qualitative analysis are shown in rectangles.

## Theme 1: Overall staff shortages in health facilities

**A. Workforce is not tailored to health facility needs.** Health care worker participants acknowledged that there were significant staffing gaps at health facilities across the district. They attributed these shortages in part to hiring policies at the central level, noting that staff distribution across different health facilities in the district was not based on the actual on-the-ground needs of a given clinic. Every clinic was allotted the same number of health care workers regardless of the volume of patients or the catchment population. There is a high expectation for the volume of work that health facilities are expected to perform. While new programs are continuously being added, there is no corresponding increase in the number of staff.

"If there is ANC [antenatal clinic] care day. . . you may find that it's not easy for you to supervise or mentor that nurse on your program [TB program] because he/she is dealing with another patient. You understand that in this case the nurse is working at ANC and also dealing with TB patients; the supervisor is not able to talk to her about the TB program while she is still busy helping patients at MCH [maternal-child health]. It is not easy to talk to the nurse because the work will be overwhelming".

[Health worker, ID# 1]

"This [allocation of human resources] was done by the Ministry of Haealth. I don't know what they were basing themselves on when they came up with it. Different clinics have different catchment areas. For example, Khubetsoana, Ha Koali, and Mahlatsa have different catchment areas, but the staffing is the same; it is like they decided that all clinics will have the same number of nurses without considering the catchment areas, performance indicators, OPD attendance, and how many people are on TB treatment. It's like the staffing is not based on our data. It's like they just decided that at the hospital, we need five doctors and 100 nurses just like that."

[Health worker, ID #2]

One consequence of understaffing is that all staff are expected to provide all forms of care, regardless of their training or affiliation with a given program. Under this work-sharing model, each staff member switches between programs on any given day to ensure that all activities in the facility are covered. The expectation that staff swap roles to meet care demands means that staff must often work without taking any breaks. Staff who complete night shifts cannot return home in the morning because nobody can take over their shift. When staff fall ill they hesitate to take sick leave because they know that this will result in a lack of coverage at the facility. The work-life balance of the staff is significantly compromised. Participants noted that this leads to burnout and negatively affects their ability to do their job.

"Because of the population we are serving, we are still understaffed. For example, the TB department is at the back in this facility; those who are already on treatment–we do not book them on daily basis, and we only book them on Tuesday and Wednesday. This is because we are short of nurses and we cannot afford to have a nurse who works only with TB patients. This is one of the restrictions we have of not having enough human resources."

[Nurse, FGD, Khubetsoana Health Center]

"I feel like each program has to have a health care worker that focuses on working on it so they don't have to work on multiple programs at the same time but still bearing in mind that they will have to go on leave whether sick leave or any other leave. They are working at night and have to be off the following day. . . and in the clinics those things are not happening. You are on call, you deliver two pregnant women, in the morning there is a long queue waiting for you outside and you have to carry on doing your work.

[Health worker, ID #2]

**B. Lack of ownership over screening activities.** Health care worker participants pointed out that no single individual was formally designated as responsible for carrying out TB screening, which is commonly performed at entry points to the health facility and in waiting areas before a patient sees their caregiver. Screening activities are shared among multiple clinic employees. At any given time, the screening could be performed by an implementing partner, a village health worker, a lay counselor, or other staff members.

Screening activities are generally allocated on a day-to-day basis, and the responsibility for screening often falls to an employee whose primary position is to support another non-TB program. While staff understand that they are required to screen when asked to do so, they do not see it as their job. Participants expressed concerned that this lack of "ownership" stemming from the shared nature of screening activities effectively eliminates the traceability of screening activities and results in an overall decline in the quality of TB screening.

"We have a challenge with the TB screening–that is why we are not getting most of the patients with TB. Because we do not have trained TB screeners, people who are doing the screening are just the facilities' employees. Moreover, they take turns in the screening of TB; if this week is lay counselor, the other week will be the VHWC [village health worker coordinator]. It is just nobody's department, so the screening is not done correctly; there is nobody who owns the screening department because no one is trained about screening. That is where the challenge is, and that is why at times when you find samples collected, nobody is feeling responsible because they will say, '[he/she was] not there that week, it was

so-and-so.' So screening is not owned by the TB screening department. If there were trained TB screeners, it would have been beneficial."

[Health worker, ID #14]

"I do it because I find it necessary to do so and because of the staff shortages. It is not in my job description, but because the VHW [village health worker] was already doing that, I had to continue screening patients."

[Health worker, ID # 6]

"There is no specific person assigned for TB screening. Anybody on duty, that is the one screening for TB, hence why it is sometimes not properly done. Few people are. . .screened at MCH, only from implementing partners."

[Nurse, FGD, Berea Hospital]

### Theme 2: Lack of adequate training for screeners

Employees from other programs who were asked to perform TB screening in addition to their work explained that they were not formally trained on how to screen for TB. They were provided with screening tools but did not receive specific training on how to use them correctly. The TB screening tools are closed-ended questionnaires that ask about key symptoms such as fever, cough, weight loss, and night sweating. These ad-hoc screeners learned about screening on the job, either through informal feedback from colleagues or observing nurses when they screened patients. Continuous refresher training was also not offered.

"I think the gap is visible when the TB screeners are not clear on what to do or how to screen TB in the facilities—especially the lay counselors. If somebody has not been trained or lacks a refresher, the results of the screening by the screener will usually be bad because he/she may miss some patients who have TB and say that they are not presumptive TB patients. So that will impact badly on TB detection."

[Health worker, ID # 1]

"The people who screen for TB in Berea hospital have never received specific training on how to screen TB. They have just been given the screening tools to screen patients for TB."

[Nurse, FGD, Berea Hospital]

"We got it [training] once or maybe two times, but usually what happens is that we have meetings as VHWs [village health workers], and during those meetings we discuss different topics and this helps us remember what we had forgotten. And when we have a problem with certain topics, we ask the nurse to help us with them."

[Health worker, ID#8]

Health care workers and the management team members were concerned that poorly trained screeners could not uniformly apply the screening tool. Furthermore, without training on how to deliver the screening tool, some participants worried that screeners were not properly explaining questions to patients. This could result in inaccurate answers to the screening question, leading to missed opportunities to identify people with symptoms, inaccurate documentation of symptoms, and compromised care. Patients with active TB could return home without a proper diagnosis.

"We miss most of the patients. Most of the TB patients are missed during the screening, and they go home with TB, which was not detected during the screening because of the way they ask questions. Patients respond, and [screeners] miss them, and they go home without being known that they have presumptive TB. Furthermore, they go home and the disease spreads to the families and communities. Later on, they come back very ill, and sometimes they are too ill even to complete the treatment and die in the process because they missed the first time they went to the facility during the screening."

[Health worker ID #14]

"The weakness may be the staff members [TB screeners] not asking the patients the screening questions in a standardized way; they ask them differently on different days."

[Health worker, ID #1]

"Because of lack of training we are missing TB cases. This will result in the rapid transmission of tuberculosis infection and this can be exacerbated with lack of knowledge on how to prevent TB infections. Lack of training also affects in the documentation of the outcome of tuberculosis treatment, particularly on the outcome of 'died' [for] TB patients."

[Nurse, FGD, Berea Hospital]

While participants acknowledge that training is essential for effective and efficient TB screening, there are simply no funds available to train staff on screening procedures.

"I think the ministry always mentions the issues of not having funds. Otherwise, every facility could choose people who could be trained for TB screening. But it is the challenge of unavailable funds for the training."

[Health worker, ID #14]

"We are actually trained by the national TB and Leprosy program, and we do the step-down trainings if we have been trained. But you may find that there are some hindering issues like hiccups with the funds when we have to decentralize the trainings."

[Health worker, ID#1]

"Like I said it is finances, human resources–these are the things we lack. So, if we could get support when it comes to these we can be very effective."

[Health worker, ID#2]

## Theme 3: Structural barriers create delays that shape patients' care priorities

**A. Long travel and clinic wait times negatively impact how patients feel when visiting a health facility.** Patients and health workers in our sample described the challenges that patients faced when attending health facilities. They had to travel long distances to the facility and often woke up early and left their homes before mealtime. Many had to walk long distances as part of their journey and often struggled to pay for transport. Upon arrival at the facility, many patients were already weak and fatigued, and they knew that they would face an additional hours-long wait before a provider could see them.

"We always queue for a long time at the gate. It brings a lot of fatigue and dizziness."

[Patient, Berea hospital, ID#7]

"We have a great problem. A person is already very sick, like I have told you that sometimes I even struggled to have food, I have already left home having not taken enough food. When you get to the facility, you get here at 8:00 am only to get consultations at 12 [noon]."

[Patient, Khubetsoana health center, ID#9]

It takes many hours [to go to the facility] because the cars [did not] come straight here [facility], as they wait for passengers.

[Patient, Berea hospital, ID#10]

"Most of the time, [it] is the distance that [hinders] the patient's accessibility of the facility. Most of the facilities are not accessible because most of the people have to travel for some distance to reach the facility."

[Health worker, ID#15]

**B. Health care workers recognize that long wait times compromise screening.** Health care workers understood that patients were focused on seeing their providers as quickly as possible to receive care and begin their challenging trip home. From their view, patients prioritized being seen by the clinician and viewed screening activities as a potential delay that added additional wait time to their already hours-long wait for care. Health care workers expressed concern that, in their view, patients provided answers to a screening question that would not prolong their time at the clinic by requiring them to visit the TB clinic for sputum collection.

"Another one is when they come here for services –just as you saw that long queue outside the gate, I think that happens in almost all clinics–there is always a long queue. People know that they are going to be screened for TB, and if they answer "yes" to one of the questions, that means you are going to take longer to get services because they make you give the sputum and wait for the results and all that."

[Health worker, ID # 2]

"The patients complain that the counselors or TB screeners make them wait too long, therefore waste their time. For example, I often overheard the patients talking on the corridors complaining that the TB screeners waste their time; therefore, the patients would rather respond 'no, no, no' on each and every screening question asked."

[Nurse, FGD, Khubetsoana Health Center]

"Once a patient has any TB symptoms, he/she is ordered to send the sputum to the TB clinic, so this makes patients unwilling to give out all the answers when being screened." [Nurse, FGD, Berea Hospital]

**Theme 4: Internal mistrust about the accuracy of screening leads to redundancy.** After a patient passes through the TB screening process, they will be able to see their provider. While screening should have already been carried out by a screener when they entered the heatlh facility, nurses reported that they do not trust the screeners' assessment. They therefore take on the additional task of of re-screening patients for TB during their clinical visit. When nurses re-screen patients during clinical consultations, this adds time to the patient visit, and also overloads nurses who are already stretched thin by their clinical duties. This creates service inefficiencies, with highly trained clinicians performing work that could be carried out by minimally trained screeners.

"Even after the counselor or TB screener has screened the patient, the nurse also screens the patient again. This is done because a patient can provide a 'no' answer to the counselor when being screened, only to admit to having such a symptom when they get to the nurses. So, we always make sure that we ask signs and symptoms of TB at every point that the patient gets to."

[Nurse, FGD, Khubetsoana Health Center]

"[We] re-screen the patient because the patient may be carrying a note that shows that he/she has been screened for TB, but if the clinician feels like re-screening the patient, they do so."

[Health worker, ID #1]

Moreover, nurses reported screening for TB in the consultation room only if they suspect TB based on physical manifestations, which means that they could be catching the disease only in later stages.

"Loss of weight is one of the skills I use to assess a presumptive TB patient. Sometimes once the patient gets into the consulting room, as a nurse, you recognize that the patient is emaciated. Sometimes just by looking at the patient's clothes, you will see that they no longer fit on the owner of the clothes. Another assessment is just the physical appearance; like when a patient has [miliary] TB, he/she will not mention night sweats. It is only when I recognize these that I will know the patient is a presumptive TB patient." [Nurse, FGD, Berea Hospital]

## Discussion

In this facility-based study, we found that poorly implemented screening procedures are likely to be a major contributor to missed TB diagnoses among people accessing the health system. We found that less than a third of visitors at the district referral hospital and around half of the visitors at the health center were asked about TB symptoms. While we do not know the reason for this difference, is possible that TB screening activities are less emphasized at the hospital based on the assumption is that patients would have already been screened at their local health center prior to referral to the hospital. Four main obstacles to effective TB screening were identified: an overall staff shortage in the health facilities, lack of adequate training for screeners and health workers in the health facilities, structural barriers create delays that shape patients' care priorities, and health care workers' mistrust of the initial screening process.

Our findings showed that the overall shortage of staff and lack of trained TB screeners contributed to the low completion of TB screening. These factors have been found to be major barriers to effective TB screening in other settings as well, contributing to poor service quality, long health service delays, low patient satisfaction, and high staff turnover [15, 16]. One potential solution to the lack of ownership over TB screening activities described by the health care workers in our study would be to introduce trained layperson or community health worker TB screeners into health facilities, which has been shown to increase TB diagnoses [6, 17]. This would require additional resources, as screeners are likely to be more effective if properly compensated [18]; however, using community health workers for activities that do not require a nurse or doctor has been shown to be cost-effective across a variety of health conditions and interventions [19]. Moreover, it would be necessary to integrate TB screening efficiently into the patients' visits so that sputum collection does not prolong the time that patients spend at the clinic.

Another potential strategy for improving TB screening in the context of understaffed health facilities is better integration of health services. In our study, health care workers mentioned that having insufficient staff to cover multiple vertical programs attending to different health conditions contributed to a lack of prioritization of TB screening–a separate activity not tied to the other programs. Integrating TB screening into standard health facility intake procedures could increase coverage with minimal additional time required [20]. Alternatively, TB screening could be incorporated as a standard evaluation procedure at every clinical encounter, as has been done successfully for people accessing HIV care in many sub-Saharan African settings [21]. Since health care workers indicated that TB screening within the context of the clinical evaluation sometimes occurs already, standardizing and monitoring the process could replace a separate screening procedure. Making screening and sputum collection part of a standard intake or evaluation procedure could potentially also reduce the disincentive for patients to respond positively to TB screening if it reduces the perception of TB screening as an additional activity that prolongs the time spent at the health facility.

While symptom-based screening in health facilities is potentially low-cost and high-yield, it is likely not sufficient to close the gap in TB detection in Lesotho or other similar settings. Known barriers to TB diagnosis in Lesotho include poor access to health care in rural areas [22] and suboptimal logistics for sputum testing [23]. Thus, efforts to improve TB detection cannot be limited to health facilities. Bringing active case-finding services into communities can help to improve TB diagnosis among people who face barriers to accessing health facilities [24]. Indeed, over half of the people with respiratory symptoms identified by the 2019 Lesotho TB prevalence survey had not sought care for their symptoms [25]. Moreover, symptom screening has serious limitations, as prolonged cough has poor sensitivity while broad symptom criteria have poor specificity [4]. Chest radiography offers high sensitivity and specificity as a screening tool, and computer-aided detection software can help to make mass radiographic screening feasible [26]. In settings with high TB burdens, community-based mobile radiography units have been used to reach populations that face barriers to accessing health facilities [27] and have been shown to increase population-level case notifications [28].

Our findings identified gaps in five of the six domains of the WHO framework for the building blocks of health systems (leadership and governance, service delivery, health informatics, health financing, human resources, and pharmacy and supplies) [29, 30]. We found deficiencies in human resources, access to quality TB services, adequate funds for training and hiring of staff, and governance and leadership in prioritizing and allocating the workforce. Moreover, the distribution of staff in the health facilities was not based on the need and patient workload, so data usage for program optimization was low. Weak health systems significantly impact TB programs [31]. Sufficient and equitable staffing, adequate supply of essential commodities, and proper monitoring and evaluation of performance through effective data utilization is critical for eliminating TB.

The study has some limitations. Firstly, we conducted the study in only in two health facilities, so our findings may not be generalizable to the rest of the country. However, by conducting interviews and focus groups with both patients and a variety of healthcare workers, our data captured a range of experiences at both the community, local facility, and district level. Also, due to COVID-19 delays in accessing data at health facilities, this study employed a convergent design rather than the originally planned sequential design, so the quantitative data identifying the significant gap in TB screening was not apparent until after the qualitative data collection was complete. As a result, we did not probe as much as we could have about patients' experiences and attitudes regarding screening, and we did not specifically recruit patients who had not been screened.

Another set of limitations stems from the paper-based data sources available for the quantitative assessment of screening coverage. To estimate the total number of visits, we assumed that each visit corresponded to a single individual eligible for screening, even though for child-focused services, each visit would involve multiple people, as children would be accompanied by adult guardians. Since cough-based screening is an adult-focused strategy, we believe that using total visits as a denominator is valid, as the total number of visits is likely to be close to the total number of adults coming to the health facility. However, this assumption reflects the lack of systematic screening of children for TB. Our assessment of the number of people screened based on screening registers is also subject to error in both directions. On the one hand, people who were screened twice–once by a screener and once during their clinical evaluation–could potentially be listed on the screening register twice. However, in practice, nurses who screen patients during the clinical evaluation often do not fill out the register. While there is thus substantial uncertainty around our estimate of screening coverage, the challenges revealed by our qualitative findings suggest that even if the registers underestimate the number of patients screened, the quality and consistency of screening is likely to be suboptimal. Finally, individual-level data extraction from paper registers was not feasible given the patient volumes of the health facilities. Thus, we were unable to collect data on important patient-level characteristics such as HIV status, which would have allowed us to assess whether TB screening differed among patient groups.

## Conclusion

To close the TB diagnosis gap in Lesotho and other countries with high TB burdens, it is essential to improve TB screening services at health facilities and in communities. Within health facilities, it is necessary to ensure that the staff conducting this activity are properly trained on both adult and pediatric TB screening, and that the coverage and quality of screening is continuously monitored. Different screening models can be considered based on the local health system context; introducing a cadre of lay worker or community health worker screeners where they do not exist could help facility-based screening programs to operate more effectively, as could better integration of TB screening into HIV, antenatal, and other primary health care services. Moreover, active TB case-finding in communities can help to remove some of the structural barriers faced by patients and thus improve in TB detection. These investments are necessary in order to find people living with undiagnosed TB and prevent the unnecessary spread of TB in the community.

## Supporting information

**S1 Text. Interview guides.**
(DOCX)

**S1 Table. Supplementary quotations.** Supporting quotes for themes identified in qualitative analysis.
(XLSX)

## Author Contributions

**Conceptualization:** Afom T. Andom, Hannah N. Gilbert, Joia S. Mukherjee, Mary C. Smith Fawzi, Courtney M. Yuen.

**Formal analysis:** Afom T. Andom, Hannah N. Gilbert, Courtney M. Yuen.

**Investigation:** Jonase Nthunya, Tholoana A. Marole.

**Project administration:** Afom T. Andom, Melino Ndayizigiye.

**Resources:** Melino Ndayizigiye, Joia S. Mukherjee.

**Supervision:** Afom T. Andom, Hannah N. Gilbert, Melino Ndayizigiye, Courtney M. Yuen.

**Writing – original draft:** Afom T. Andom, Hannah N. Gilbert, Courtney M. Yuen.

**Writing – review & editing:** Afom T. Andom, Hannah N. Gilbert, Melino Ndayizigiye, Joia S. Mukherjee, Jonase Nthunya, Tholoana A. Marole, Mary C. Smith Fawzi, Courtney M. Yuen.

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
