## [Decision Letter · Decision Letter 0]

26 Oct 2021

PGPH-D-21-00687

Understanding reasons for low tuberculosis detection in a low-resource setting: a mixed-methods study in the Kingdom of Lesotho

Dear Dr. Yuen,

Thank you for submitting your manuscript to PLOS Global Public Health. After careful consideration, we feel that it has merit but does not fully meet PLOS Global Public Health’s publication criteria as it currently stands. Therefore, we invite you to submit a revised version of the manuscript that addresses the points raised during the review process.

We look forward to receiving your revised manuscript.

Kind regards,

Megan Coffee, MD, PhD

Academic Editor

Additional Editor Comments (if provided):

Thank you very much for your submission of your work on "Understanding reasons for low tuberculosis detection in a low-resource setting: a mixed-methods study in the Kingdom of Lesotho". I think the topic and approach are precisely what PLOS Global Health is looking to publish. You will see there are many thoughtful comments and suggestions to ensure the conclusions match the study findings. I hope you find these helpful.

There also are some minor corrections to ensure uniformity and precision in language, though the paper itself is very readable. Thank you and I hope we are able to go forward with this paper, guided by the suggestions included.

Reviewers' comments:

Reviewer's Responses to Questions

**Comments to the Author**

1. Does this manuscript meet PLOS Global Public Health’s publication criteria? Is the manuscript technically sound, and do the data support the conclusions? The manuscript must describe methodologically and ethically rigorous research with conclusions that are appropriately drawn based on the data presented.

Reviewer #1: Yes

Reviewer #2: Yes

2. Has the statistical analysis been performed appropriately and rigorously?

Reviewer #1: Yes

Reviewer #2: Yes

3. Have the authors made all data underlying the findings in their manuscript fully available (please refer to the Data Availability Statement at the start of the manuscript PDF file)?

Reviewer #1: No

Reviewer #2: Yes

4. Is the manuscript presented in an intelligible fashion and written in standard English?

Reviewer #1: Yes

Reviewer #2: Yes

5. Review Comments to the Author

Reviewer #1: This paper is on an important topic from a high-burden setting and is an important piece of work utilizing a clever methodology. There are major problems, however, with the manuscript that need to be addressed prior to publication.

First, and perhaps most importantly, the paper is really an exploration as to why the rates for TB SCREENING are low, not as to why the rates of TB detection are low. While it is great that the authors present all steps of the cascade from the quantitative analysis, the fact that they only explore the reasons behind low rates of screening in the qualitative assessment really means their paper should focus on this aspect of the cascade. Thus I would suggest that they reframe the entire paper in this way. This should start with the title--which should be changed to state it is TB SCREENING and not DETECTION that will be explored--and carry throughout the paper. I think this might help the authors focus the background and discussion more as well. And while I think it is worth presenting the whole cascade with its results, the authors really should try and target the paper more on TB screening. To this end, it would help to have something in the background about the health care resources/services available in Lesotho as well as other experiences with implementing TB symptom screening systematically (for example this has been done in South Africa for quite some time). I also think it would be important in the discussion section to talk about the limitations of TB symptom screening as part of the care cascade and how it misses many people with active TB, even when done systematically. I find it interesting, for example, that the authors note in their own introduction section in lines 41-43 that screening should preferably done with chest X-ray, thus suggesting they also see a limit to TB symptom screening. But since the bulk of the paper is about why TB symptom screening is not done, I would suggest the authors substantially revise the paper to focus on this. I suspect this will take a couple of rounds of revisions with reviewers, but I would encourage the authors to try and do this if they are willing as there are some rich and important findings in their study.

If the authors want to focus on other aspects of the care cascade, then there are some other findings that merit discussion that are completely unaddressed in their mixed methods. Their rates of sputum collection and submission are strikingly high as are their rates of treatment initiation, and this should be commented on if the authors chose to explore other aspects of the cascade. However, their rates of treatment completion are unacceptably low and this should also be explored. It was not clear to me if authors actually collected qualitative data on these aspects of the cascade and simply chose not to present them for reasons of space/economy or of the only collected qualitative data on why people did not undergo screening. If they have qualitative data on these other aspects of the cascade, they could certainly consider presenting them in a revised manuscript--although I suspect it could get quite lengthy.

My second major concern with the paper is that there seem to be some internal inconsistencies with the data shared in the qualitative assessments and the data in the quantitative cascade. This happens often in mixed methods research, as people are often inconsistent with reporting or understanding the motivations for their behaviors. But when this happens it must be addressed and explored in the discussion section. One example of this is the reported qualitative finding that nurses did not "trust" the screeners to really do the TB symptom screening and thus they re-screened the patients themselves. But if this is true--then it would mean that people were screened TWICE at the health facilities (once by screeners and once by nurses) and rather a higher rate of symptom screening should be reported (unless this is, of course, not really happening or not being "recorded" in a way that was captured in the data analysis). Another example is saying that patients have to wait for long periods of time when they come to the clinic but that they do not have time to undergo TB screening. If they are queued and waiting for a long time, could the screening not also be done then? During the early days of antiretroviral availability in Lesotho, for example, the long waiting times were used to do HIV education and counseling, as people were queuing for hours to wait for physicians. While it may be that people did not want to talk about TB or get "pulled" from the queue to do further TB testing, this should lead to a discussion on the need to better streamline TB screening and testing procedures for people. There are other instances of this throughout the paper that must be addressed in the discussion and limitations section, as some of what people appear to be reporting does not match what the numbers show. This is one of the beautiful things about mixed methods studies in that you have different information coming from different actors/sources. But these things must be discussed.

There is a different in the percentage of people screened for TB symptoms at the hospital versus the health center and the authors should comment on this in the discussion and present any qualitative data they might have that could illuminate the difference. One would expect a hospital to be better staffed and resourced than a health center--although perhaps they were also overwhelmed with too many critically ill patients? Whatever the reason, this should be noted and explored.

The interview guide should be included as an Annex.

There should be a COREQ checklist used to report on this qualitative study and that should be submitted with the revision and noted in the methods.

Was there any attempt in the estimates of number of people who should be screened for TB to account or adjust for TB screening in children, where different symptoms might suggest TB? Where sputum may be difficult to obtain? This should be mentioned since the authors note they used data from the vaccine and under 5 clinics to calculate the number of people who should have been screened for TB.

In line 278-279, the authors note that the "lack of specialization" may be one reason why TB screening was not done. I am not sure if this is their term of if this is what participants gave as a reason. It is worth exploring this further in the discussion as well, since one "goal" of a PHC is that everyone can do multiple jobs--and it is precisely this lack of "specialization" that is the goal in an integrated, holistic system. Screening for TB symptoms is not a complex intervention where "specialization" is needed. So how would this fit in with a whole UHC goal--which I know Lesotho has invested a lot in--as a way to improve TB care? Again, if this is what participants stated as a reason for not doing TB screening, then the authors should report it this way. But this is an important finding that then has implications for trying to get all PHC staff or community health workers to be able to do TB screening and shows the global move towards universal screening may not match with what people on the ground are seeing or want ("specialized people" to do screening).

Some of the language needs to be more scientific and less informal. For example, the test done to look for TB is a cartridge-based genotypic test and not a "GeneXpert" machine (line 45), it is not clear what a "mild" symptom is (line 42). The group that transports the sputum specimens is called Riders for Health (not Riders on Health, line 82). There is also some stigmatizing language used in the paper that should be addressed (i.e. while it is a common epidemiology term to refer to "TB cases" these are people who are living with TB--line 62).

I realize I am suggesting a major re-write/reconceptualization of this piece of work, but I would encourae the authors to consider it if they can. I think since the bulk of the qualitative data are presented on why people are not doing screening, then the paper should be revised around that. If the authors do want to focus on the whole cascade, then they need to present qualitative data on the other aspects of the cascade, because some of the steps are completed at a higher rate than would be expected (sputum collection, initiation in treatment) and others are also much lower than expected (treatment completion).

Reviewer #2: Authors have made a good attempt on the subject matter. Issues requiring attention:

1. Authors need to ensure consistency in the use of British and American, in the abstract the authors have used Supervised the British way and Incentivized the American way

2. The conclusion of the study is not supported by the study findings. The authors have indicated in the conclusion that case finding could improve if health care workers are supervised. I did not see any data supporting this conclusion

3.In the discussion, the authors need to relate their findings with others studies. They made an attempt but not consistently.

6. PLOS authors have the option to publish the peer review history of their article (what does this mean?). If published, this will include your full peer review and any attached files.

**Do you want your identity to be public for this peer review?** For information about this choice, including consent withdrawal, please see our Privacy Policy.

Reviewer #1: No

Reviewer #2: No

---

## [Decision Letter · Decision Letter 1]

17 Jan 2022

PGPH-D-21-00687R1

Understanding reasons for suboptimal tuberculosis screening in a low-resource setting: a mixed-methods study in the Kingdom of Lesotho

Dear Dr. Yuen,

Thank you for submitting your manuscript to PLOS Global Public Health. After careful consideration, we feel that your paper meets the criteria for publishing except for one small question detailed in the reviewer section, which we would like to ask a minor revision for. Do submit this and we will act right away on this.

We look forward to receiving your revised manuscript.

Kind regards,

Megan Coffee, MD, PhD

Academic Editor

Journal Requirements:

Additional Editor Comments (if provided):

Thank you very much for your submission. It has been a busy time for everyone in infectious diseases. We appreciate your patience and all the work you did to complete the revisions requested. We are looking forward to publishing this. If you are able to make the small revision or provide feedback on "visitors" raised by one of the reviewers, that is the only outstanding issue. We can move along quickly without sending out to further reviewers, just do reach back about the "visitors" comment.

Reviewers' comments:

Reviewer's Responses to Questions

**Comments to the Author**

1. If the authors have adequately addressed your comments raised in a previous round of review and you feel that this manuscript is now acceptable for publication, you may indicate that here to bypass the “Comments to the Author” section, enter your conflict of interest statement in the “Confidential to Editor” section, and submit your "Accept" recommendation.

Reviewer #1: All comments have been addressed

Reviewer #2: All comments have been addressed

2. Does this manuscript meet PLOS Global Public Health’s publication criteria? Is the manuscript technically sound, and do the data support the conclusions? The manuscript must describe methodologically and ethically rigorous research with conclusions that are appropriately drawn based on the data presented.

Reviewer #1: Yes

Reviewer #2: Yes

3. Has the statistical analysis been performed appropriately and rigorously?

Reviewer #1: Yes

Reviewer #2: Yes

4. Have the authors made all data underlying the findings in their manuscript fully available (please refer to the Data Availability Statement at the start of the manuscript PDF file)?

Reviewer #1: Yes

Reviewer #2: Yes

5. Is the manuscript presented in an intelligible fashion and written in standard English?

Reviewer #1: Yes

Reviewer #2: Yes

6. Review Comments to the Author

Reviewer #1: The authors have done an excellent job revising this paper, and they have addressed all the comments made by me and by the other reviewer. I realize this involved a substantial amount of work and re-analysis, and I am impressed at how thoughtfully and thoroughly the authors were able to update the paper. The work is much stronger now and makes a compelling piece of evidence on one aspect of the TB care cascade that is failing.

I have one very minor comment to make. The authors use the term "visitors" to the health facilities, and I understand this to mean people who are coming there to seek health care. However, when referring to the hospital, the term visitors could also mean people who are there to visit family members or other people who are sick in the hospital. Are these people also screened for TB? Should they be? Or do the authors mean to use the term "visitors" to convey people who come there seeking care? I realize this is minor, but the term should probably be clarified somewhere in the paper (i.e. a small sentence saying "We use the term "visitor" to denote people who came to the facility seeking health services" if this is in fact what is meant. I do note that the authors state that family members often brought children for care and those family members are screened as well, so maybe they do mean the term to encompass anyone entering the facility for any reason, and if so, this should be mentioned.).

Thank you for taking the comments into account and doing such a nice round of revisions.

Reviewer #2: The authors have addressed most the issues that were raised in the previous submission and the manuscript reads well. They may wish to correct a few issues:

1. In the abstract they have used TB screeners, this is not an official term in reference to the health care workers involved in screening for TB. The author may consider using the term clinical team

2. In line 83 the term once found positive for TB symptoms, can be substituted with TB presumptive

3. In line 108 - 112 the authors need to observe the space before the = sign

4. in line 477 I suggest that for the reference 29 and 30 they use close the square brackets for each reference as [29], [30]

Otherwise well done to the team

7. PLOS authors have the option to publish the peer review history of their article (what does this mean?). If published, this will include your full peer review and any attached files.

**Do you want your identity to be public for this peer review?** For information about this choice, including consent withdrawal, please see our Privacy Policy.

Reviewer #1: No

Reviewer #2: **Yes: **Patrick Lungu

---

## [Editor Report · Decision Letter 2]

13 Feb 2022

Understanding reasons for suboptimal tuberculosis screening in a low-resource setting: a mixed-methods study in the Kingdom of Lesotho

PGPH-D-21-00687R2

Dear Dr Yuen

We are pleased to inform you that your manuscript 'Understanding reasons for suboptimal tuberculosis screening in a low-resource setting: a mixed-methods study in the Kingdom of Lesotho' has been provisionally accepted for publication in PLOS Global Public Health.

Best regards,

Megan Coffee, MD, PhD

Academic Editor